# Likelihood Ratios for Out-of-Distribution Detection

**Jie Ren**[*][†]
Google Research
jjren@google.com

**Peter J. Liu** [‡]
Google Research
peterjliu@google.com

**Emily Fertig**[†]
Google Research
emilyaf@google.com

**Jasper Snoek**
Google Research
jsnoek@google.com

**Ryan Poplin**
Google Research
rpoplin@google.com

**Mark A. DePristo**
Google Research
mdepristo@google.com

**Joshua V. Dillon** [‡]
Google Research
jvdillon@google.com

**Balaji Lakshminarayanan**[*][‡]
DeepMind
balajiln@google.com

## Abstract

Discriminative neural networks offer little or no performance guarantees when deployed on data not generated by the same process as the training distribution. On such out-of-distribution (OOD) inputs, the prediction may not only be erroneous, but confidently so, limiting the safe deployment of classifiers in real-world applications. One such challenging application is bacteria identification based on genomic sequences, which holds the promise of early detection of diseases, but requires a model that can output low confidence predictions on OOD genomic sequences from new bacteria that were not present in the training data. We introduce a genomics dataset for OOD detection that allows other researchers to benchmark progress on this important problem. We investigate deep generative model based approaches for OOD detection and observe that the likelihood score is heavily affected by population level background statistics. We propose a likelihood ratio method for deep generative models which effectively corrects for these confounding background statistics. We benchmark the OOD detection performance of the proposed method against existing approaches on the genomics dataset and show that our method achieves state-of-the-art performance. We demonstrate the generality of the proposed method by showing that it significantly improves OOD detection when applied to deep generative models of images.

## 1 Introduction

For many machine learning systems, being able to detect data that is anomalous or significantly different from that used in training can be critical to maintaining safe and reliable predictions. This is particularly important for deep neural network classifiers which have been shown to incorrectly classify such *out-of-distribution* (OOD) inputs into in-distribution classes with high confidence (Goodfellow et al., 2014; Nguyen et al., 2015). This behaviour can have serious consequences when the predictions inform real-world decisions such as medical diagnosis, e.g. falsely classifying a healthy sample as pathogenic or vice versa can have extremely high cost. The importance of dealing with OOD inputs, also referred to as distributional shift, has been recognized as an important problem for

---

[*]Corresponding authors

[†]Google AI Resident

[‡]Mentors

AI safety (Amodei et al., 2016). The majority of recent work on OOD detection for neural networks is evaluated on image datasets where the neural network is trained on one benchmark dataset (e.g. CIFAR-10) and tested on another (e.g. SVHN). While these benchmarks are important, there is a need for more realistic datasets which reflect the challenges of dealing with OOD inputs in practical applications.

Bacterial identification is one of the most important sub-problems of many types of medical diagnosis. For example, diagnosis and treatment of infectious diseases, such as sepsis, relies on the accurate detection of bacterial infections in blood (Blauwkamp et al., 2019). Several machine learning methods have been developed to perform bacteria identification by classifying existing known genomic sequences (Patil et al., 2011; Rosen et al., 2010), including deep learning methods (Busia et al., 2018) which are state-of-the-art. Even if neural network classifiers achieve high accuracy as measured through cross-validation, deploying them is challenging as real data is highly likely to contain genomes from unseen classes not present in the training data. Different bacterial classes continue to be discovered gradually over the years (see Figure S4 in Appendix C.1) and it is estimated that 60%-80% of genomic sequences belong to as yet unknown bacteria (Zhu et al., 2018; Eckburg et al., 2005; Nayfach et al., 2019). Training a classifier on existing bacterial classes and deploying it may result in OOD inputs being wrongly classified as one of the classes from the training data with high confidence. In addition, OOD inputs can also be the contaminations from the bacteria's host genomes such as human, plant, fungi, etc., which also need to be detected and excluded from predictions (Ponsero & Hurwitz, 2019). Thus having a method for accurately detecting OOD inputs is critical to enable the practical application of machine learning methods to this important problem.

A popular and intuitive strategy for detecting OOD inputs is to train a generative model (or a hybrid model cf. Nalisnick et al. (2019)) on training data and use that to detect OOD inputs at test time (Bishop, 1994). However, Nalisnick et al. (2018) and Choi et al. (2018) recently showed that deep generative models trained on image datasets can assign higher likelihood to OOD inputs. We report a similar failure mode for likelihood based OOD detection using deep generative models of genomic sequences. We investigate this phenomenon and find that the likelihood can be confounded by general population level background statistics. We propose a likelihood ratio method which uses a background model to correct for the background statistics and enhances the in-distribution specific features for OOD detection. While our investigation was motivated by the genomics problem, we found our methodology to be more general and it shows positive results on image datasets as well. In summary, our contributions are:

- We create a realistic benchmark for OOD detection, that is motivated by challenges faced in applying deep learning models on genomics data. The sequential nature of genetic sequences provides a new modality and hopefully encourages the OOD research community to contribute to "machine learning that matters" (Wagstaff, 2012).

- We show that likelihood from deep generative models can be confounded by background statistics.

- We propose a likelihood ratio method for OOD detection, which significantly outperforms the raw likelihood on OOD detection for deep generative models on image datasets.

- We evaluate existing OOD methods on the proposed genomics benchmark and demonstrate that our method achieves state-of-the-art (SOTA) performance on this challenging problem.

## 2    Background

Suppose we have an in-distribution dataset $\mathcal{D}$ of $(\boldsymbol{x}, y)$ pairs sampled from the distribution $p^*(\boldsymbol{x}, y)$, where $\boldsymbol{x}$ is the extracted feature vector or raw input and $y \in \mathcal{Y} := \{1, \ldots, k, \ldots, K\}$ is the label assigning membership to one of $K$ in-distribution classes. For simplicity, we assume inputs to be discrete, i.e. $x_d \in \{A, C, G, T\}$ for genomic sequences and $x_d \in \{0, \ldots, 255\}$ for images. In general, OOD inputs are samples $(\boldsymbol{x}, y)$ generated from an underlying distribution other than $p^*(\boldsymbol{x}, y)$. In this paper, we consider an input $(\boldsymbol{x}, y)$ to be OOD if $y \notin \mathcal{Y}$: that is, the class $y$ does not belong to one of the $K$ in-distribution classes. Our goal is to accurately detect if an input $\boldsymbol{x}$ is OOD or not.

Many existing methods involve computing statistics using the predictions of (ensembles of) discriminative classifiers trained on in-distribution data, e.g. taking the confidence or entropy of the predictive distribution $p(y|\boldsymbol{x})$ (Hendrycks & Gimpel, 2016; Lakshminarayanan et al., 2017). An alternative is to use generative model-based methods, which are appealing as they do not require

labeled data and directly model the input distribution. These methods fit a generative model $p(\boldsymbol{x})$ to the input data, and then evaluate the likelihood of new inputs under that model. However, recent work has highlighted significant issues with this approach for OOD detection on images, showing that deep generative models such as Glow (Kingma & Dhariwal, 2018) and PixelCNN (Oord et al., 2016; Salimans et al., 2017) sometimes assign higher likelihoods to OOD than in-distribution inputs. For example, Nalisnick et al. (2018) and Choi et al. (2018) show that Glow models trained on the CIFAR-10 image dataset assign higher likelihood to OOD inputs from the SVHN dataset than they do to in-distribution CIFAR-10 inputs; Nalisnick et al. (2018), Shafaei et al. (2018) and Hendrycks et al. (2018) show failure modes of PixelCNN and PixelCNN++ for OOD detection.

**Failure of density estimation for OOD detection** We investigate whether density estimation-based methods work well for OOD detection in genomics. As a motivating observation, we train a deep generative model, more precisely LSTM (Hochreiter & Schmidhuber, 1997), on in-distribution genomic sequences (composed by {A, C, G, T}), and plot the log-likelihoods of both in-distribution and OOD inputs (See Section 5.2 for the dataset and the full experimental details). Figure 1a shows that the histogram of the log-likelihood for OOD sequences largely overlaps with that of in-distribution sequences with AUROC of 0.626, making it unsuitable for OOD detection. Our observations show a failure mode of deep generative models for OOD detection on genomic sequences and are complementary to earlier work which showed similar results for deep generative models on images (Nalisnick et al., 2018; Choi et al., 2018).

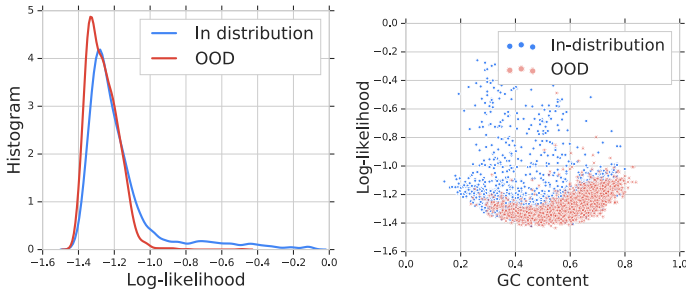

Figure 1: (a) Log-likelihood hardly separates in-distribution and OOD inputs with AUROC of 0.626. (b) The log-likelihood is heavily affected by the GC-content of a sequence.

When investigating this failure mode, we discovered that the log-likelihood under the model is heavily affected by a sequence's *GC-content*, see Figure 1b. GC-content is defined as the percentage of bases that are either G or C, and is used widely in genomic studies as a basic statistic for describing overall genomic composition (Sueoka, 1962), and studies have shown that bacteria have an astonishing diversity of genomic GC-content, from 16.5% to 75% (Hildebrand et al., 2010). Bacteria from similar groups tend to have similar GC-content at the population level, but they also have characteristic biological patterns that can distinguish them well from each other. The confounding effect of GC-content in Figure 1b makes the likelihood less reliable as a score for OOD detection, because an OOD input may result in a higher likelihood than an in-distribution input, because it has high GC-content (cf. the bottom right part of Figure 1b) and not necessarily because it contains characteristic patterns specific to the in-distribution bacterial classes.

## 3   Likelihood Ratio for OOD detection

We first describe the high level idea and then describe how to adapt it to deep generative models.

**High level idea** Assume that an input $\boldsymbol{x}$ is composed of two components, (1) a *background* component characterized by population level background statistics, and (2) a *semantic* component characterized by patterns specific to the in-distribution data. For example, images can be modeled as backgrounds plus objects; text can be considered as a combination of high frequency stop words plus semantic words (Luhn, 1960); genomes can be modeled as background sequences plus motifs (Bailey & Elkan, 1995; Reinert et al., 2009). More formally, for a $D$-dimensional input $\boldsymbol{x} = x_1, \ldots, x_D$, we assume that there exists an unobserved variable $\boldsymbol{z} = z_1, \ldots, z_D$, where $z_d \in \{B, S\}$ indicates if the $d$th dimension of the input $x_d$ is generated from the $B$ackground model or the $S$emantic model.

Grouping the semantic and background parts, the input can be factored as $\boldsymbol{x} = \{\boldsymbol{x}_B, \boldsymbol{x}_S\}$ where $\boldsymbol{x}_B = \{x_d \mid z_d = B, d = 1, \ldots, D\}$. For simplicity, assume that the background and semantic components are generated independently. The likelihood can be then decomposed as follows,

$$p(\boldsymbol{x}) = p(\boldsymbol{x}_B)p(\boldsymbol{x}_S). \tag{1}$$

When training and evaluating deep generative models, we typically do not distinguish between these two terms in the likelihood. However, we may want to use just the semantic likelihood $p(\boldsymbol{x}_S)$ to avoid the likelihood term being dominated by the background term (e.g. OOD input with the same background but different semantic component). In practice, we only observe $\boldsymbol{x}$, and it is not always easy to split an input into background and semantic parts $\{\boldsymbol{x}_B, \boldsymbol{x}_S\}$. As a practical alternative, we propose training a background model by perturbing inputs. Adding the right amount of perturbations to inputs can corrupt the semantic structure in the data, and hence the model trained on perturbed inputs captures only the population level background statistics.

Assume that $p_{\boldsymbol{\theta}}(\cdot)$ is a model trained using in-distribution data, and $p_{\boldsymbol{\theta}_0}(\cdot)$ is a background model that captures general background statistics. We propose a likelihood ratio statistic that is defined as

$$\mathsf{LLR}(\boldsymbol{x}) = \log \frac{p_{\boldsymbol{\theta}}(\boldsymbol{x})}{p_{\boldsymbol{\theta}_0}(\boldsymbol{x})} = \log \frac{p_{\boldsymbol{\theta}}(\boldsymbol{x}_B)\, p_{\boldsymbol{\theta}}(\boldsymbol{x}_S)}{p_{\boldsymbol{\theta}_0}(\boldsymbol{x}_B)\, p_{\boldsymbol{\theta}_0}(\boldsymbol{x}_S)}, \tag{2}$$

where we use the factorization from Equation 1. Assume that (i) both models capture the background information equally well, that is $p_{\boldsymbol{\theta}}(\boldsymbol{x}_B) \approx p_{\boldsymbol{\theta}_0}(\boldsymbol{x}_B)$ and (ii) $p_{\boldsymbol{\theta}}(\boldsymbol{x}_S)$ is more peaky than $p_{\boldsymbol{\theta}_0}(\boldsymbol{x}_S)$ as the former is trained on data containing semantic information, while the latter model $\boldsymbol{\theta}_0$ is trained using data with noise perturbations. Then, the likelihood ratio can be approximated as

$$\mathsf{LLR}(\boldsymbol{x}) \approx \log p_{\boldsymbol{\theta}}(\boldsymbol{x}_S) - \log p_{\boldsymbol{\theta}_0}(\boldsymbol{x}_S). \tag{3}$$

After taking the ratio, the likelihood for the background component $\boldsymbol{x}_B$ is cancelled out, and only the likelihood for the semantic component $\boldsymbol{x}_S$ remains. Our method produces a *background contrastive* score that captures the significance of the semantics compared with the background model.

**Likelihood ratio for auto-regressive models** Auto-regressive models are one of the popular choices for generating images (Oord et al., 2016; Van den Oord et al., 2016; Salimans et al., 2017) and sequence data such as genomics (Zou et al., 2018; Killoran et al., 2017) and drug molecules (Olivecrona et al., 2017; Gupta et al., 2018), and text (Jozefowicz et al., 2016). In auto-regressive models, the log-likelihood of an input can be expressed as $\log p_{\boldsymbol{\theta}}(\boldsymbol{x}) = \sum_{d=1}^{D} \log p_{\boldsymbol{\theta}}(x_d|\boldsymbol{x}_{<d})$, where $\boldsymbol{x}_{<d} = x_1 \ldots x_{d-1}$. Decomposing the log-likelihood into background and semantic parts, we have

$$\log p_{\boldsymbol{\theta}}(\boldsymbol{x}) = \sum_{d:x_d \in \boldsymbol{x}_B} \log p_{\boldsymbol{\theta}}(x_d|\boldsymbol{x}_{<d}) + \sum_{d:x_d \in \boldsymbol{x}_S} \log p_{\boldsymbol{\theta}}(x_d|\boldsymbol{x}_{<d}). \tag{4}$$

We can use a similar auto-regressive decomposition for the background model $p_{\boldsymbol{\theta}_0}(\boldsymbol{x})$ as well. Assuming that both the models capture the background information equally well, $\sum_{d:x_d \in \boldsymbol{x}_B} \log p_{\boldsymbol{\theta}}(x_d|\boldsymbol{x}_{<d}) \approx \sum_{d:x_d \in \boldsymbol{x}_B} \log p_{\boldsymbol{\theta}_0}(x_d|\boldsymbol{x}_{<d})$, the likelihood ratio is approximated as

$$\mathsf{LLR}(\boldsymbol{x}) \approx \sum_{d:x_d \in \boldsymbol{x}_S} \log p_{\boldsymbol{\theta}}(x_d|\boldsymbol{x}_{<d}) - \sum_{d:x_d \in \boldsymbol{x}_S} \log p_{\boldsymbol{\theta}_0}(x_d|\boldsymbol{x}_{<d}) = \sum_{d:x_d \in \boldsymbol{x}_S} \log \frac{p_{\boldsymbol{\theta}}(x_d|\boldsymbol{x}_{<d})}{p_{\boldsymbol{\theta}_0}(x_d|\boldsymbol{x}_{<d})}. \tag{5}$$

**Training the Background Model** In practice, we add perturbations to the input data by randomly selecting positions in $x_1 \ldots x_D$ following an independent and identical Bernoulli distribution with rate $\mu$ and substituting the original character with one of the other characters with equal probability. The procedure is inspired by genetic mutations. See Algorithm 1 in Appendix A for the pseudocode for generating input perturbations. The rate $\mu$ is a hyperparameter and can be easily tuned using a small amount of validation OOD dataset (different from the actual OOD dataset of interest). In the case where validation OOD dataset is not available, we show that $\mu$ can also be tuned using simulated OOD data. In practice, we observe that $\mu \in [0.1, 0.2]$ achieves good performance empirically for most of the experiments in our paper. Besides adding perturbations to the input data, we found other techniques that can improve model generalization and prevent model memorization, such as adding $L_2$ regularization with coefficient $\lambda$ to model weights, can help to train a good background model. In fact, it has been shown that adding noise to the input is equivalent to adding $L_2$ regularization to the model weights under some conditions (Bishop, 1995a,b). Besides the methods above, we expect adding other types of noise or regularization methods would show a similar effect. The pseudocode for our proposed OOD detection algorithm can be found in Algorithm 2 in Appendix A.

# 4 Experimental setup

We design experiments on multiple data modalities (images, genomic sequences) to evaluate our method and compare with other baseline methods. For each of the datasets, we build an autoregressive model for computing the log-likelihood $\log p_{\boldsymbol{\theta}}(\boldsymbol{x}) = \sum_{d=1}^{D} \log p_{\boldsymbol{\theta}}(x_d|\boldsymbol{x}_{<d})$. For training the background model $p_{\boldsymbol{\theta}_0}(\boldsymbol{x})$, we use the exact same architecture as $p_{\boldsymbol{\theta}}(\boldsymbol{x})$, and the only differences are that it is trained on perturbed inputs and (optionally) we apply $L_2$ regularization to model weights.

**Baseline methods for comparison** We compare our approach to several existing methods.

1. The maximum class probability, $p(\hat{y}|\boldsymbol{x}) = \max_k p(y = k|\boldsymbol{x})$. OOD inputs tend to have lower scores than in-distribution data (Hendrycks & Gimpel, 2016).

2. The entropy of the predicted class distribution, $-\sum_k p(y = k|\boldsymbol{x}) \log p(y = k|\boldsymbol{x})$. High entropy of the predicted class distribution, and therefore a high predictive uncertainty, which suggests that the input may be OOD.

3. The ODIN method proposed by Liang et al. (2017). ODIN uses temperature scaling (Guo et al., 2017), adds small perturbations to the input, and applies a threshold to the resulting predicted class to distinguish in- and out-of- distribution inputs. This method was designed for continuous inputs and cannot be directly applied to discrete genomic sequences. We propose instead to add perturbations to the input of the last layer that is closest to the output of the neural network.

4. The Mahalanobis distance of the input to the nearest class-conditional Gaussian distribution estimated from the in-distribution data. Lee et al. (2018) fit class-conditional Gaussian distributions to the activations from the last layer of the neural network.

5. The classifier-based ensemble method that uses the average of the predictions from multiple independently trained models with random initialization of network parameters and random shuffling of training inputs (Lakshminarayanan et al., 2017).

6. The log-odds of a binary classifier trained to distinguish between in-distribution inputs from all classes as one class and randomly perturbed in-distribution inputs as the other.

7. The maximum class probability over $K$ in-distribution classes of a $(K+1)$-class classifier where the additional class is perturbed in-distribution.

8. The maximum class probability of a $K$-class classifier for in-distribution classes but the predicted class distribution is explicitly trained to output uniform distribution on perturbed in-distribution inputs. This is similar to using simulated OOD inputs from GAN (Lee et al., 2017) or using auxiliary datasets of outliers (Hendrycks et al., 2018) for calibration purpose.

9. The generative model-based ensemble method that measures $\mathbb{E}[\log p_{\boldsymbol{\theta}}(\boldsymbol{x})] - \mathrm{Var}[\log p_{\boldsymbol{\theta}}(\boldsymbol{x})]$ of multiple likelihood estimations from independently trained model with random initialization and random shuffling of the inputs. (Choi et al., 2018).

Baseline methods 1-8 are classifier-based and method 9 is generative model-based. For classifier-based methods, we choose the commonly used model architecture, convolutional neural networks (CNNs). Methods 6-8 are based on perturbed inputs which aims to mimic OOD inputs. Perturbations are added to the input in the same way as that we use for training background models. Our method and methods 3, 6, 7, and 8 involve hyperparameter tuning; we follow the protocol of Hendrycks et al. (2018) where optimal hyperparameters are picked on a different OOD validation set than the final OOD dataset it is tested on. For Fashion-MNIST vs. MNIST experiment, we use the NotMNIST Bulatov (2011) dataset for hyperparameter tuning. For CIFAR-10 vs SVHN, we used gray-scaled CIFAR-10 for hyperparameter tuning. For genomics, we use the OOD bacteria classes discovered between 2011-2016, which are disjoint from the final OOD classes discovered after 2016. While this set of baselines is not exhaustive, it is broadly representative of the range of existing methods. Note that since our method does not rely on OOD inputs for training, we do not compare it with other methods that do utilize OOD inputs in training.

**Evaluation metrics for OOD detection** We trained the model using only in-distribution inputs, and we tuned the hyperparameters using validation datasets that include both in-distribution and OOD inputs. The test dataset is used for the final evaluation of the method. For the final evaluation, we randomly selected the same number of in-distribution and OOD inputs from the test dataset, and for each example $\boldsymbol{x}$ we computed the log likelihood-ratio statistic LLR($\boldsymbol{x}$) as the score. A small value of

the score suggests a high likelihood of being OOD. We use the area under the ROC curve (AUROC↑), the area under the precision-recall curve (AUPRC↑), and the false positive rate at 80% true positive rate (FPR80↓), as the metrics for evaluation. These three metrics are commonly used for evaluating OOD detection methods (Hendrycks & Gimpel, 2016; Hendrycks et al., 2018; Alemi et al., 2018).

# 5 Results

We first present results on image datasets as they are easier to visualize, and then present results on our proposed genomic dataset. For image experiments, our goal is not to achieve state-of-the-art performance but to show that our likelihood ratio effectively corrects for background statistics and significantly outperforms the likelihood. While previous work has shown the failure of PixelCNN for OOD detection, we believe ours is the first to provide an explanation for why this phenomenon happens for PixelCNN, through the lens of background statistics.

## 5.1 Likelihood ratio for detecting OOD images

Following existing literature (Nalisnick et al., 2018; Hendrycks et al., 2018), we evaluate our method using two experiments for detecting OOD images: (a) Fashion-MNIST as in-distribution and MNIST as OOD, (b) CIFAR-10 as in-distribution and SVHN as OOD. For each experiment, we train a PixelCNN++ (Salimans et al., 2017; Van den Oord et al., 2016) model using in-distribution data. We train a background model by adding perturbations to the training data. To compare with classifier-based baseline methods, we use CNN-based classifiers. See Appendix B.1 for model details. Based on the likelihood from the PixelCNN++ model, we confirm that the model assigns a higher likelihood to MNIST than Fashion-MNIST, as previously reported by Nalisnick et al. (2018), and the AUROC for OOD detection is only 0.091, even worse than random (Figure 2a). We discover that the proportion of zeros i.e. *number of pixels belonging to the background in an image is a confounding factor to the likelihood score* (Pearson Correlation Coefficient 0.85, see Figure 2b, Figure S1). Taking the likelihood ratio between the original and the background models, we see that the AUROC improves significantly from 0.091 to 0.996 (Figure 2d). The log likelihood-ratio for OOD images are highly concentrated around value 0, while that for in-distribution images are mostly positive (Figure 2c).

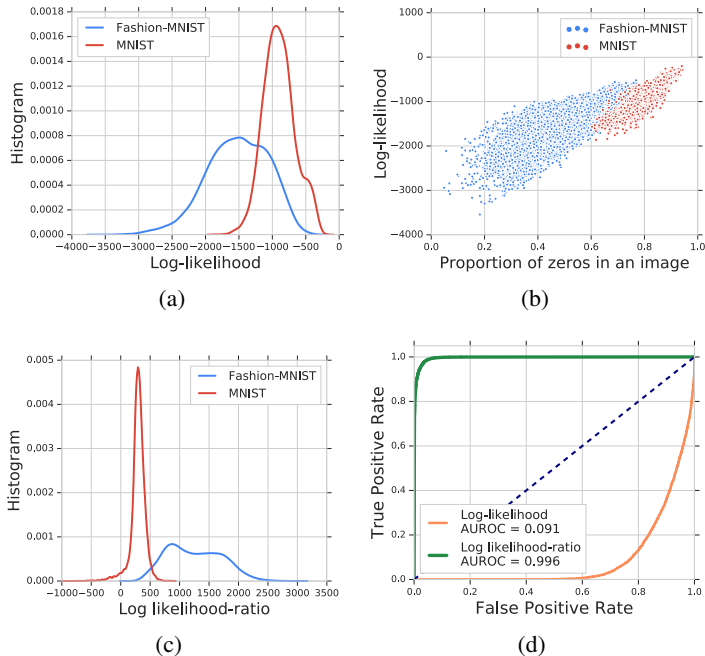

(a)  (b)

(c)  (d)

Figure 2: (a) Log-likelihood of MNIST images (OOD) is higher than that of Fashion-MNIST images (in-distribution). (b) Log-likelihood is highly correlated with the background (proportion of zeros in an image). (c) Log-likelihood ratio is higher for Fashion-MNIST (in-dist) than MNIST (OOD). (d) Likelihood ratio significantly improves the AUROC of OOD detection from 0.091 to 0.996.

**Which pixels contribute the most to the likelihood (ratio)?** To qualitatively evaluate the difference between the likelihood and the likelihood ratio, we plot their values for each pixel for Fashion-MNIST and MNIST images. This allows us to visualize which pixels contribute the most to the two terms respectively. Figure 3 shows a heatmap, with lighter (darker) gray colors indicating higher (lower) values. Figures 3(a,b) show that the likelihood value is dominated by the "background" pixels, whereas likelihood ratio focuses on the "semantic" pixels. Figures 3(c,d) confirm that the background pixels cause MNIST images to be assigned high likelihood, whereas likelihood ratio focuses on the semantic pixels. We present additional qualitative results in Appendix B. For instance, Figure S2 shows that images with the highest likelihood-ratios are those with prototypical Fashion-MNIST icons, e.g. "shirts" and "bags", highly contrastive with the background, while images with the lowest likelihood-ratios are those with rare patterns, e.g. dress with stripes or sandals with high ropes.

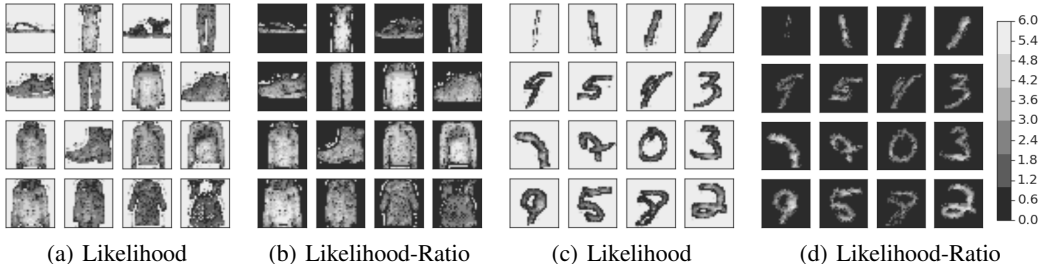

(a) Likelihood      (b) Likelihood-Ratio      (c) Likelihood      (d) Likelihood-Ratio

Figure 3: The log-likelihood of each pixel in an image $\log p_{\boldsymbol{\theta}}(x_d|\boldsymbol{x}_{<d})$, and the log likelihood-ratio of each pixel $\log p_{\boldsymbol{\theta}}(x_d|\boldsymbol{x}_{<d}) - \log p_{\boldsymbol{\theta}_0}(x_d|\boldsymbol{x}_{<d})$, $d = 1 \ldots, 784$., for 16 Fashion-MNIST images (a, b) and MNIST images (c, d). Lighter gray color indicates larger value (see colorbar). Note that the range of log-likelihood (negative value) is different from that of log likelihood-ratio (mostly positive value). For the ease of visualization, we unify the colorbar by adding a constant to the log-likelihood score. The images are randomly sampled from the test dataset and sorted by their likelihood $p_{\theta}(\mathbf{x})$. Looking at which pixels contribute the most to each quantity, we observe that the likelihood value is dominated by the "background" pixels on both Fashion-MNIST and MNIST, whereas likelihood ratio focuses on the "semantic" pixels.

We compare our method with other baselines. The classifier-based baseline methods are built using LeNet architecture. Table 1a shows that our method achieves the highest AUROC↑, AUPRC↑, and the lowest FPR80↓. The method using Mahalanobis distance performs better than other baselines. Note that the binary classifier between in-distribution and perturbed in-distribution does not perform as well as our method, possibly due to the fact that while the features learned by the discriminator can be good for detecting perturbed inputs, they may not generalize well for OOD detection. The generative model approach based on $p(\boldsymbol{x})$ captures more fundamental features of the data generation process than the discriminative approach.

For the experiment using CIFAR-10 as in-distribution and SVHN as OOD, we apply the same training procedure using the PixelCNN++ model and choose hyperparameters using grayscaled CIFAR-10 which was shown to be OOD by Nalisnick et al. (2018). See Appendix B.1 for model details. Looking at the results in Table 2, we observe that the OOD images from SVHN have higher likelihood than the in-distribution images from CIFAR-10, confirming the observations of Nalisnick et al. (2018), with AUROC of 0.095. Our likelihood-ratio method significantly improves the AUROC to 0.931. Figure S3 in Appendix B shows additonal qualitative results. For detailed results including other baseline methods, see Table S2 in Appendix B.3.

## 5.2 OOD detection for genomic sequences

**Dataset for detecting OOD genomic sequences** We design a new dataset for evaluating OOD methods. As bacterial classes are discovered gradually over time, in- and out-of-distribution data can be naturally separated by the time of discovery. Classes discovered before a cutoff time can be regarded as in-distribution classes, and those discovered afterward, which were unidentified at the cutoff time, can be regarded as OOD. We choose two cutoff years, 2011 and 2016, to define the training, validation, and test splits (Figure 4). Our dataset contains of 10 in-distribution classes, 60 OOD classes for validation, and 60 OOD classes for testing. Note that the validation OOD dataset is only used for hyperparameter tuning, and the validation OOD classes are disjoint from the test OOD

Table 1: AUROC↑, AUPRC↑, and FPR80↓ for detecting OOD inputs using likelihood and likelihood-ratio method and other baselines on (a) Fashion-MNIST vs. MNIST datasets and (b) genomic dataset. The up and down arrows on the metric names indicate whether greater or smaller is better. $\mu$ in the parentheses indicates the background model is tuned only using noise perturbed input, and ($\mu$ and $\lambda$) indicates the background model is tuned by both perturbation and $L_2$ regularization. Numbers in front and inside of the brackets are mean and standard error respectively based on 10 independent runs with random initialization of network parameters and random shuffling of training inputs. For ensemble models, the mean and standard error are estimated based on 10 bootstrap samples from 30 independent runs, which can be underestimations of the true standard errors.

(a)

| | AUROC↑ | AUPRC↑ | FPR80↓ |
|---|---|---|---|
| Likelihood | 0.089 (0.002) | 0.320 (0.000) | 1.000 (0.001) |
| Likelihood Ratio (ours, $\mu$) | 0.973 (0.031) | 0.951 (0.063) | 0.005 (0.008) |
| Likelihood Ratio (ours, $\mu, \lambda$) | **0.994 (0.001)** | **0.993 (0.002)** | **0.001 (0.000)** |
| $p(\hat{y}|\boldsymbol{x})$ | 0.734 (0.028) | 0.702 (0.026) | 0.506 (0.046) |
| Entropy of $p(y|\boldsymbol{x})$ | 0.746 (0.027) | 0.726 (0.026) | 0.448 (0.049) |
| ODIN | 0.752 (0.069) | 0.763 (0.062) | 0.432 (0.116) |
| Mahalanobis distance | 0.942 (0.017) | 0.928 (0.021) | 0.088 (0.028) |
| Ensemble, 5 classifiers | 0.839 (0.010) | 0.833 (0.009) | 0.275 (0.019) |
| Ensemble, 10 classifiers | 0.851 (0.007) | 0.844 (0.006) | 0.241 (0.014) |
| Ensemble, 20 classifiers | 0.857 (0.005) | 0.849 (0.004) | 0.240 (0.011) |
| Binary classifier | 0.455 (0.105) | 0.505 (0.064) | 0.886 (0.126) |
| $p(\hat{y}|\boldsymbol{x})$ with noise class | 0.877 (0.050) | 0.871 (0.054) | 0.195 (0.101) |
| $p(\hat{y}|\boldsymbol{x})$ with calibrations | 0.904 (0.023) | 0.895 (0.023) | 0.139 (0.044) |
| WAIC, 5 models | 0.221 (0.013) | 0.401 (0.008) | 0.911 (0.008) |

(b)

| | AUROC↑ | AUPRC↑ | FPR80↓ |
|---|---|---|---|
| Likelihood | 0.626 (0.001) | 0.613 (0.001) | 0.661 (0.002) |
| Likelihood Ratio (ours, $\mu$) | 0.732 (0.015) | 0.685 (0.017) | 0.534 (0.031) |
| Likelihood Ratio (ours, $\mu, \lambda$) | **0.755 (0.005)** | **0.719 (0.006)** | **0.474 (0.011)** |
| $p(\hat{y}|\boldsymbol{x})$ | 0.634 (0.003) | 0.599 (0.003) | 0.669 (0.007) |
| Entropy of $p(y|\boldsymbol{x})$ | 0.634 (0.003) | 0.599 (0.003) | 0.617 (0.007) |
| Adjusted ODIN | 0.697 (0.010) | 0.671 (0.012) | 0.550 (0.021) |
| Mahalanobis distance | 0.525 (0.010) | 0.503 (0.007) | 0.747 (0.014) |
| Ensemble, 5 classifiers | 0.682 (0.002) | 0.647 (0.002) | 0.589 (0.004) |
| Ensemble, 10 classifiers | 0.690 (0.001) | 0.655 (0.002) | 0.574 (0.004) |
| Ensemble, 20 classifiers | 0.695 (0.001) | 0.659 (0.001) | 0.570 (0.004) |
| Binary classifier | 0.635 (0.016) | 0.634 (0.015) | 0.619 (0.025) |
| $p(\hat{y}|\boldsymbol{x})$ with noise class | 0.652 (0.004) | 0.627 (0.005) | 0.643 (0.008) |
| $p(\hat{y}|\boldsymbol{x})$ with calibration | 0.669 (0.005) | 0.635 (0.004) | 0.627 (0.006) |
| WAIC, 5 models | 0.628 (0.001) | 0.616 (0.001) | 0.657 (0.002) |

Table 2: CIFAR-10 vs SVHN results: AUROC↑, AUPRC↑, FPR80↓ for detecting OOD inputs using likelihood and our likelihood-ratio method.

| | AUROC↑ | AUPRC↑ | FPR80↓ |
|---|---|---|---|
| Likelihood | 0.095 (0.003) | 0.320 (0.001) | 1.000 (0.000) |
| Likelihood Ratio (ours, $\mu$) | 0.931 (0.032) | 0.888 (0.049) | 0.062 (0.073) |
| Likelihood Ratio (ours, $\mu, \lambda$) | 0.930 (0.042) | 0.881 (0.064) | 0.066 (0.123) |

classes. To mimic sequencing data, we fragmented genomes in each class into short sequences of 250 base pairs, which is a common length that current sequencing technology generates. Among all the short sequences, we randomly choose 100,000 sequences for each class for training, validation, and test. Additional details about the dataset, including pre-processing and the information for the in- and out-of-distribution classes, can be found in Appendix C.1.

**Likelihood ratio method for detecting OOD sequences** We build an LSTM model for estimating the likelihood $p(\boldsymbol{x})$ based on the transition probabilities $p(x_d|\boldsymbol{x}_{<d})$, $d = 1, \ldots, D$. In particular, we feed the one-hot encoded DNA sequences into an LSTM layer, followed by a dense layer and a softmax function to predict the probability distribution over the 4 letters of $\{A, C, G, T\}$, and train the model using only the in-distribution training data. We evaluate the likelihood for sequences in the OOD test dataset under the trained model, and compare those with the likelihood for sequences in the in-distribution test dataset. The AUROC↑, AUPRC↑, and FPR80↓ scores are 0.626, 0.613, and 0.661 respectively (Table 1b).

We train a background model by using the perturbed in-distribution data and optionally adding $L_2$ regularization to the model weights. Hyperparameters are tuned using validation dataset which contains in-distribution and validation OOD classes, and the validation OOD classes are disjoint from test OOD classes. Contrasting against the background model, the AUROC↑, AUPRC↑, and FPR80↓ for the likelihood-ratio significantly improve to 0.755, 0.719, and 0.474, respectively (Table 1b, Figure 5b). Compared with the likelihood, the AUROC and AUPRC for likelihood-ratio increased 20% and 17% respectively, and the FPR80 decreased 28%. Furthermore, Figure 5a shows that the likelihood ratio is less sensitive to GC-content, and the separation between in-distribution and OOD distribution becomes clearer. We evaluate other baseline methods on the test dataset as well. For classifier-based baselines, we construct CNNs with one convolutional layer, one max-pooling layer, one dense layer, and a final dense layer with softmax activation for predicting class probabilities, as in Alipanahi et al. (2015); Busia et al. (2018); Ren et al. (2018b). Comparing our method to the baselines in Table 1b, our method achieves the highest AUROC, AUPRC, and the lowest FPR80 scores on the test dataset. Ensemble method and ODIN perform better than other baseline methods. Comparing with the Fashion-MNIST and MNIST experiment, the Mahalanobis distance performs worse for detecting genomic OOD possibly due to the fact that Fashion-MNIST and MNIST images are quite distinct while in-distribution and OOD bacteria classes are interlaced under the same taxonomy (See Figure S5 for the phylogenetic tree of the in-distribution and OOD classes).

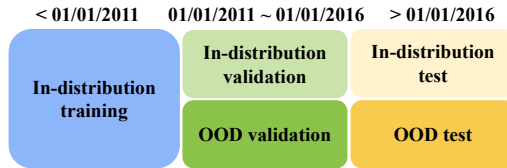

Figure 4: The design of the training, validation, and test datasets for genomic sequence classification including in and OOD data.

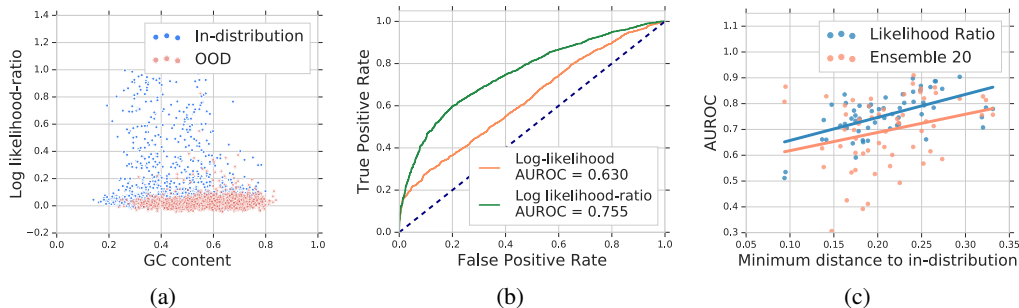

| (a) | (b) | (c) |

Figure 5: (a) The likelihood-ratio score is roughly independent of the GC-content which makes it less susceptible to background statistics and better suited for OOD detection. (b) ROCs and AUROCs for OOD detection using likelihood and likelihood-ratio. (c) Correlation between the AUROC of OOD detection and distance to in-distribution classes using Likelihood Ratio and the Ensemble method.

**OOD detection correlates with its distance to in-distribution** We investigate the effect of the distance between the OOD class to the in-distribution classes, on the performance of OOD detection. To measure the distance between the OOD class to the in-distribution, we randomly select representative genome from each of the in-distribution classes and OOD classes. We use the state-of-the-art alignment-free method for genome comparison, $d_2^S$ (Ren et al., 2018a; Reinert et al., 2009), to compute the genetic distance between each pair of the genomes in the set. This genetic distance is calculated based on the similarity between the normalized nucleotide word frequencies ($k$-tuples) of the two genomes, and studies have shown that this genetic distance reflects true evolutionary distances between genomes (Chan et al., 2014; Bernard et al., 2016; Lu et al., 2017). For each of the OOD classes, we use the minimum distance between the genome in that class to all genomes in the in-distribution classes as the measure of the genetic distance between this OOD class and the in-distribution. Not surprisingly, the the AUROC for OOD detection is positively correlated with the genetic distance (Figure 5c), and an OOD class far away from in-distribution is easier to be detected. Comparing our likelihood ratio method and one of the best classifier-based methods, ensemble method, we observe that our likelihood ratio method has higher AUROC for different OOD classes than ensemble method in general. Furthermore, our method has a higher Pearson correlation coefficient (PCC) of 0.570 between the minimum distance and AUROC for Likelihood Ratio method, than the classifier-based ensemble method with 20 models which has PCC of 0.277. The dataset and code for the genomics study is available at https://github.com/google-research/google-research/tree/master/genomics_ood.

## 6   Discussion and Conclusion

We investigate deep generative model-based methods for OOD detection and show that the likelihood of auto-regressive models can be confounded by background statistics, providing an explanation to the failure of PixelCNN for OOD detection observed by recent work (Nalisnick et al., 2018; Hendrycks et al., 2018; Shafaei et al., 2018). We propose a likelihood ratio method that alleviates this issue by contrasting the likelihood against a background model. We show that our method effectively corrects for the background components, and significantly improves the accuracy of OOD detection on both image datasets and genomic datasets. Finally, we create and release a realistic genomic sequence dataset for OOD detection which highlights an important real-world problem, and hope that this serves as a valuable OOD detection benchmark for the research community.

**Acknowledgments**

We thank Alexander A. Alemi, Andreea Gane, Brian Lee, D. Sculley, Eric Jang, Jacob Burnim, Katherine Lee, Matthew D. Hoffman, Noah Fiedel, Rif A. Saurous, Suman Ravuri, Thomas Colthurst, Yaniv Ovadia, the Google Brain Genomics team, and Google TensorFlow Probability team for helpful feedback and discussions.

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
