[Supplementary Material]

# Appendix

## A  Additional details about our proposed likelihood ratio method

The pseudocode for generating input perturbations for training the background model is described in Algorithm 1.

---
**Algorithm 1** Input perturbation for background model training
---
1: **Inputs**: $D$-dimensional input $\boldsymbol{x} = x_1 \ldots x_D$, Mutation rate $\mu$ and vocabulary $\mathcal{A}$. Note that we assume inputs to be discrete, i.e. $x_d \in \mathcal{A}$, where $\mathcal{A} = \{A, C, G, T\}$ for genomic sequences and $\mathcal{A} = \{0, \ldots, 255\}$ for images.
2: **Output**: perturbed input $\tilde{\boldsymbol{x}}$
3: Generate a $D$-dimensional vector $\mathbf{v} = v_1 \ldots, v_D$, where $v_d \in \{0, 1\}$ are independent and identically distributed according to a Bernoulli distribution with rate $\mu$.
4: **for** index $d \in \{1, \ldots, D\}$ **do**
5:     **if** $v_d = 1$ **then**
6:         Sample $\tilde{x}_d$ from the set $\mathcal{A}$ with equal probability.
7:     **else**
8:         Set $\tilde{x}_d = x_d$.
9:     **end if**
10: **end for**

---

The complete pseudocode for our method is described in Algorithm 2. The runtime of our method is two times of the standard generative model runtime.

---
**Algorithm 2** OOD detection using Likelihood Ratio
---
1: Fit a model $p_{\boldsymbol{\theta}}(\boldsymbol{x})$ using in-distribution dataset $\mathcal{D}_{\text{in}}$.
2: Fit a background model $p_{\boldsymbol{\theta}_0}(\boldsymbol{x})$ using perturbed input data $\widetilde{\mathcal{D}}_{\text{in}}$ (generated using Algorithm 1) and (optionally) model regularization techniques.
3: Compute the likelihood ratio statistic $\text{LLR}(\boldsymbol{x})$
4: Predict OOD if $\text{LLR}(\boldsymbol{x})$ is small.

---

## B  Supplementary materials for the experiments on images

### B.1  Model details

Following existing literature (Nalisnick et al., 2018; Choi et al., 2018), we evaluate our method using two experiments for detecting OOD images: (a) Fashion-MNIST as in-distribution and MNIST as OOD, (b) CIFAR-10 as in-distribution and SVHN as OOD. For the experiment of Fashion-MNIST vs. MNIST, we train a generative model using the training set of Fashion-MNIST, and use the test set of Fashion-MNIST and the test set of MNIST as OOD as the final test dataset. The same rule is applied for CIFAR-10 vs. SVHN experiment. Since SVHN test set has more inputs than CIFAR-10 test set, we randomly select the same number of inputs for evalaution.

For each experiment, we train a PixelCNN++ (Salimans et al., 2017; Van den Oord et al., 2016) model on the in-distribution data using maximum likelihood. For Fashion-MNIST dataset, the model uses 2 blocks of 5 gated ResNet layers with 32 convolutional 2D filters (concatenate ELU activation function, and without weight normalization and dropout), and 1 component in the logitstic mixture, and is trained for 50,000 steps with initial learning rate of 0.0001 with exponential decay rate of 0.999995 per step, batch size of 32, and Adam optimizer with momentum parameter $\beta_1 = 0.95$ and $\beta_2 = 0.9995$. For CIFAR-10 dataset, the model uses 2 blocks of 5 gated ResNet layers with 160 convolutional 2D filters, and 10 components in the logitstic mixture, and is trained for 600,000 steps with the same learning rate, batch size, and optimizer as the above. The bits per dimension of the

PixelCNN++ models are 2.92 and 1.64 for in-distribution images Fashion-MNIST and OOD images MNIST respectively, and 3.20 and 2.15 for in-distribution images CIFAR-10 and OOD images SVHN, respectively.

For the background model, we train a PixelCNN++ model with the same architecture on perturbed inputs obtained by randomly flipping input pixel values to one of the 256 possible values following an independent and identical Bernoulli distribution with rate $\mu$ (see Algorithm 1). The mutation rate $\mu$ for adding perturbations to input in the background model training is a hyperparameter to our method. For tuning the hyperparameters, we use independent datasets, NotMNIST (Bulatov, 2011) for Fashion-MNIST vs. MNIST experiment, and gray-scaled CIFAR-10 for CIFAR-10 vs. SVHN experiment, as the validation OOD dataset. We choose the optimal mutation rate $\mu$ based on the AUROC for OOD detection using the validation dataset. We test the mutation rate $\mu$ from the range of $[0.1, 0.2, 0.3]$. For Fashion-MNIST vs. MNIST, the optimal mutation rate tuned using NotMNIST as OOD dataset is $\mu = 0.3$ for most of the 10 independent runs. We also add $L_2$ regularization to model weights. We let the $L_2$ coefficient $\lambda$ ranges from $\lambda = [0, 10, 100]$ and test different combinations of $\mu$ and $\lambda$ on the background model training. We find that adding $L_2$ regularization helps to improve the final test AUROC only slightly from 0.973 to 0.996. The optimal combination is $\mu = 0.3$ and $\lambda = 100$ for most of the 10 independent runs of random initialization of network parameters and random shuffling of training inputs. Table S1a shows one of those, which results in AUROC of 0.996 in the final test dataset. For CIFAR-10 vs. SVHN experiment, we observe that tuning for both mutation rate and $L_2$ regularization achieves similar performance with tuning for mutation rate only. The optimal mutation rate is $\mu = 0.1$ for most of the 10 independent runs (data not shown).

In the case where no independent OOD data (such as NotMNIST) is available for hyperparameter tuning, we can use randomly mutated in-distribution input at mutation rate 2%, to mimic the OOD input. The mutation is added using the same procedure as that for training the background model. The optimal hyperparameter setting is $\mu = 0.1$ and $\lambda = 100$, which achieves AUROC of 0.958 in the final test dataset. The results show that the hyperparameters for the background model are easy to tune. Under the situation where indepedent OOD data are not available, using only simulated OOD data we are able to achieve reasonable performance.

We found that the performance of our method LLR can be slightly affected by different PixelCNN++ network setups. We tested both versions of PixelCNN++ where input images ranging from 0 to 255 are (a) directly fed into the networks (b) re-scaled to the range of -1 to 1 and fed into the networks. The two setups give different initializatiobs of the network. We found that the version without rescaling achieves slightly better performance in terms of AUROC possibly because it learns a better background model based on its initialization. The AUROC↑, AUPRC↑, and FPR80↓ for Fashion-MNIST vs. MNIST in Table 1a are based on the version without rescaling. Using the version with rescaling, the numbers for the three evaluation metrics are 0.936 (0.003), 0.891 (0.004), 0.025 (0.003), without changing model parameters (the same 5 gated ResNet with 32 convolutional 2D filters, batch size, learning rate, optimizer as before), and training for 600,000 steps. For CIFAR-10 vs. SVHN experiment, we found rescaling helps to produce more stable results. So we report AUROC↑, AUPRC↑, and FPR80↓ based on the version with rescaling in Table S2.

Table S1: Hyperparameter tuning of mutation rate $\mu$ and $L_2$ coefficient $\lambda$ of the background model of our likelihood ratio method for Fashion-MNIST vs. MNIST experiment. (a) AUROC is evaluated based on in-distribution Fashion-MNIST validation dataset and NotMNIST dataset. Note that MNIST is not used for hyperparameter turning. (b) The same as (a) but tuning using simulated OOD inputs without exposing to any NotMNIST or MNIST images. The simulated OOD inputs are generated by permuting the in-distribution inputs at the mutation rate 2%.

|  | $\mu = 0.1$ | 0.2 | 0.3 |  |  | $\mu = 0.1$ | 0.2 | 0.3 |
|---|---|---|---|---|---|---|---|---|
| $\lambda = 0$ | 0.358 | 0.426 | **0.795** |  | $\lambda = 0$ | **0.984** | 0.971 | 0.856 |
| 10 | 0.433 | 0.432 | 0.489 |  | 10 | **0.989** | 0.977 | 0.971 |
| 100 | 0.414 | 0.777 | **0.798** |  | 100 | **0.989** | 0.851 | 0.857 |

To compare with classifier-based baseline methods, we build convolutional neural networks (CNNs). We used LeNet (LeCun et al., 1998) architecture for Fashion-MNIST images. The model composes two convolutional layers with 32 and 64 2D filters respectively of size 3 by 3 with ReLU activation function, a max pooling layers of size 2 by 2, a dropout layer with the rate of 0.25, a dense layer of

128 units with ReLU activation function, another dropout layer with the rate of 0.25 based on the flattened output from the previous layer, and a final dense layer with the softmax activation function used for generating the class probabilities. Model parameters are trained for 12 epochs with batch size of 128, learning rate of 0.002, and Adam optimizer. The prediction accuracy on test data is 0.923 for in-distribution Fashion-MNIST images. For CIFAR-10 images, we build ResNet-20 V1 architecture with ReLU activations (He et al., 2016). Model parameters are trained for 120 epochs with batch size of 7, Adam optimizer, and a learning rate schedule that multiplied an initial learning rate of 0.0007 by 0.1, 0.01, 0.001, and 0.0005 at steps 80, 120, 160, and 180 respectively. Training inputs are randomly distorted using horizontal flips and random crops preceded by 4-pixel padding as described in He et al. (2016). The prediction accuracy on test data is 0.911 for in-distribution CIFAR-10 images. Both LeNet and ResNet-20 V1 model architectures are consistent with those in Ovadia et al. (2019).

For the baseline methods 6-8 that are based on perturbed inputs, the perturbation rate is tuned from the range of $[10^{-5}, 10^{-4}, \dots, 10^{-1}]$ based on validation in-distribution dataset and an independent dataset that is different from the final OOD test dataset, e.g. NotMNIST for Fashion-MNIST vs. MNIST experiment, and grayscaled CIFAR-10 for CIFAR-10 vs. SVHN experiment. Similarly the hyperparameters in ODIN method, the temperature scaling to logits and perturbations to inputs, are tuned based on the same validation dataset. The temperature is tuned in the range of $[1, 5, 10, 100, 1000]$, and the perturbation is tuned in the range of $[0, 10^{-8}, 10^{-7}, \dots, 10^{-1}]$.

## B.2 Supplementary figures

Images with the highest and lowest likelihood in Fashion-MNIST and MNIST dataset are shown in Figure S1. Images with the highest likelihoods are mostly "sandals" in Fashion-MNIST dataset and "1"s in MNIST dataset that have a large proportion of zeros. Images with the highest and lowest likelihood-ratio are shown in Figure S2. Images with the highest likelihood-ratios are those with prototypical Fashion-MNIST icons such as "shirts" and "bags", highly contrastive with the background, while images with the lowest likelihood-ratios are those with rare patterns, such as dress with stripes or sandals with high ropes.

Figure S3 shows qualitative results on CIFAR-10, displaying the per-pixel likelihood and likelihood-ratio as a heatmap. Similar to the trends in Figure 3, we observe that "background" pixels cause some CIFAR-10 images to be assigned high likelihood.

(a) FashionMNIST: highest log-likelihood.  (b) FashionMNIST: lowest log-likelihood.  (c) MNIST: highest log-likelihood.  (d) MNIST: lowest log-likelihood.

Figure S1: FashionMNIST images with (a) the highest log-likelihood, and (b) the lowest log-likelihood. MNIST images with (c) the highest log-likelihood, and (d) the lowest log-likelihood.

## B.3 Supplementary tables

The AUROC↑, AUPRC↑, and FPR80↓ for CIFAR-10 (in-distribution) vs. SVHN (OOD) experiment is shown in Table S2. Pure likelihood performs poorly for OOD detection with AUROC 0.095, and likelihood-ratio improves the performance significantly, achieving AUROC 0.931. Classifier-based ensemble methods perform well with AUROCs ranging from 0.937 to 0.946. Note that our likelihood-ratio method is completely unsupervised whereas classifier methods require labels. Using likelihood-ratios on class-conditional generative models might further improve performance.

(a) FashionMNIST: highest log-likelihood ratio.

(b) FashionMNIST: lowest log-likelihood ratio.

Figure S2: FashionMNIST images with (a) the highest and (b) the lowest log-likelihood-ratios.

(a)  (b)  (c)

Figure S3: Examples of CIFAR-10 images (a), and their corresponding log-likelihood of each pixel in an image $\log p_{\boldsymbol{\theta}}(x_d|\boldsymbol{x}_{<d})$ (b), and the log likelihood-ratio of each pixel $\log p_{\boldsymbol{\theta}}(x_d|\boldsymbol{x}_{<d}) - \log p_{\boldsymbol{\theta}_0}(x_d|\boldsymbol{x}_{<d})$, are plotted as $32 \times 32$ images. Lighter (darker) gray color indicates larger (smaller) value. Note that the range of log-likelihood (negative value) is different from that of log likelihood-ratio (mostly positive value). For the ease of visualization, we unify the colorbar by adding a constant to the log-likelihood score. We picked the images that have the highest log-likelihood $p_{\boldsymbol{\theta}}(\mathbf{x})$.

Table S2: AUROC↑, AUPRC↑, FPR80↓ for detecting OOD inputs using likelihood and likelihood-ratio method and other baselines on CIFAR-10 vs. SVHN datasets. Note that the generative-model based approaches (including our LLR method) are completely unsupervised, which puts them at a slight disadvantage, compared to classifier-based methods which have access to labels. Numbers in front and inside of the brackets are mean and standard error respectively based on 10 independent runs with random initialization of network parameters and random shuffling of training inputs. For ensemble models, mean and standard error are estimated based on 10 bootstrap samples from 30 independent runs, which can be underestimations of the true standard errors.

| | AUROC↑ | AUPRC↑ | FPR80↓ |
|---|---|---|---|
| Likelihood | 0.095 (0.003) | 0.320 (0.001) | 1.000 (0.000) |
| Likelihood Ratio (ours, $\mu$) | 0.931 (0.032) | 0.888 (0.049) | 0.062 (0.073) |
| Likelihood Ratio (ours, $\mu, \lambda$) | 0.930 (0.042) | 0.881 (0.064) | 0.066 (0.123) |
| $p(\hat{y}|\boldsymbol{x})$ | 0.910 (0.011) | 0.871 (0.019) | 0.094 (0.030) |
| Entropy of $p(\hat{y}|\boldsymbol{x})$ | 0.920 (0.013) | 0.890 (0.021) | 0.139 (0.020) |
| ODIN | 0.938 (0.091) | 0.926 (0.103) | 0.086 (0.179) |
| Mahalanobis distance | 0.728 (0.108) | 0.711 (0.118) | 0.469 (0.178) |
| Ensemble, 5 classifiers | 0.937 (0.010) | 0.906 (0.017) | 0.037 (0.013) |
| Ensemble, 10 classifiers | 0.943 (0.004) | 0.915 (0.008) | 0.023 (0.002) |
| Ensemble, 20 classifiers | 0.946 (0.002) | 0.916 (0.004) | 0.020 (0.001) |
| Binary classifier | 0.508 (0.027) | 0.505 (0.021) | 0.948 (0.107) |
| $p(y|x)$ with noise class | 0.923 (0.011) | 0.892 (0.016) | 0.064 (0.026) |
| $p(y|x)$ with calibration | 0.809 (0.043) | 0.735 (0.046) | 0.396 (0.101) |
| WAIC, 5 models | 0.146 (0.089) | 0.343 (0.038) | 0.956 (0.062) |

# C Supplementary materials for the experiments on genomic sequences

## C.1 Dataset design

We downloaded 11,672 bacteria genomes from National Center for Biotechnology Information (NCBI, https://www.ncbi.nlm.nih.gov/genome/browse#!/prokaryotes/) on September 2018. For each genome we pooled its taxonomy information from the species level, to the genus, the family, the order, the class, and the phylum level. High taxonomy levels such as the phylum level represents broad classification, while low taxonomy levels like the species and genus give a refined classification. To provide a precise classification, we use different genera as class labels, as has been done in previous studies (Brady & Salzberg, 2009; Ahlgren et al., 2016). We filtered genomes that have missing genus information, or have ambiguous genus names. A genus usually contains genomes from different species, subspecies, or strains.

Figure S4: Cumulative number of new bacteria classes discovered over the years (NCBI microbial genomes browser, September 2018.)

Different bacterial classes were discovered gradually over the years (Figure S4). Grouping classes by years is a natural way to mimic the in-distribution and OOD examples. Given a certain cutoff year, the classes discovered before the year cutoff can be regarded as in-distribution classes, and those after the year cutoff can be regarded as the OODs. In particular, we define the year that a class was first discovered as the earliest year when any of the genomes belonging to this class was discovered. We choose two cutoff years, 2011 and 2016, to define the training dataset for in-distribution, validation datasets for in-distribution and OOD, and test datasets for in-distribution and OOD (Figure 4). Genomes belonging to classes that were first discovered before 01/01/2011 are used for generating the training dataset for in-distribution. Genomes belonging to new classes that were first discovered between 01/01/2011 and 01/01/2016 are used for generating the validation dataset for OOD. Genomes belonging to the old classes but sequenced and released between 01/01/2011 and 01/01/2016 are used for generating the validation dataset for in-distribution. Similarly, genomes belonging to the new classes that were first discovered after 01/01/2016 are used for generating test dataset for OOD, while genomes belonging to the old classes that were sequenced and released after 01/01/2016 are used for generating the test dataset for in-distribution. This setting avoids overlaps among genomes from training, validation, and test datasets. It is possible that different bacteria genomes may share similar gene regions, but those are rare for genomes from different genera and hence, we ignore this effect in our study. The bacteria class names are listed in Table S3.

We designed a dataset containing 10 in-distribution classes, 60 OOD classes for validation, and 60 OOD classes for test. The classes were chosen since they are the most common classes and have the largest sample sizes. The in-distribution and OOD classes are interlaced under the same taxonomy (Figure S5). To mimic the real sequencing data, we fragmented genomes in each class into short sequences of length 250 base pairs, which is a common length that the current sequencing technology generates. Among all the short sequences, we randomly choose 100,000 sequences for each class for the training, validation, and test datasets.

Figure S5: The phylogenetic tree of the 10 in-distribution, 60 OOD validation, and 60 OOD test bacteria classes. Note that the in-distribution and OOD classes are interlaced under the same taxonomy.

## C.2 Model details

For generative models of genomic sequences, we build a LSTM model (Hochreiter & Schmidhuber, 1997) to estimate the probability distribution of the next position given the history $p(x_d|\boldsymbol{x}_{<d})$. In particular, we feed the one-hot encoded DNA sequences into an LSTM layer, followed by a dense layer and a softmax function to predict the probability distribution over the 4 letters of $\{A, C, G, T\}$. The model is trained using the in-distribution training data only. The size of the hidden layer in the LSTM was tuned via the in-distribution validation dataset and the final model uses 2,000 hidden units. The model is trained for 900,000 steps using learning rate of 0.0005, batch size of 100, and Adam optimizer. The accuracy for predicting next character is 0.45 for in-distribution inputs.

We train a background model by using the perturbed in-distribution data and (optionally) adding $L_2$ regularization to model weights. We search the optimal mutation rate $\mu$ from the range of $\mu = [0.01, 0.05, 0.1, 0.2]$, and evaluate the AUROC of 2,000 in-distribution and the same number of OOD inputs in the validation dataset. Note that the set of OOD classes in the validation dataset is different

Table S3: The bacterial classes used in the genomic dataset for in- and out-of- distributions.

| | |
|---|---|
| In-distribution training | Bacillus, Burkholderia, Clostridium, Escherichia, Mycobacterium, Pseudomonas, Salmonella, Staphylococcus, Streptococcus, Yersinia |
| OOD validation | Actinoplanes, Advenella, Alicycliphilus, Altererythrobacter, Anabaena, Archangium, Bibersteinia, Blastochloris, Calothrix, Carnobacterium, Cedecea, Cellulophaga, Chondromyces, Chryseobacterium, Collimonas, Corallococcus, Cyclobacterium, Dehalobacter, Desulfosporosinus, Devosia, Dyella, Elizabethkingia, Glaciecola, Granulicella, Haliscomenobacter, Hymenobacter, Kibdelosporangium, Kutzneria, Labilithrix, Leptolyngbya, Leptospirillum, Lysobacter, Mannheimia, Massilia, Methanobacterium, Microbacterium, Myroides, Neorhizobium, Niastella, Oblitimonas, Octadecabacter, Oscillatoria, Pandoraea, Pelosinus, Phaeobacter, Piscirickettsia, Planococcus, Pseudonocardia, Pseudoxanthomonas, Rahnella, Raoultella, Rufibacter, Saccharothrix, Sandaracinus, Singulisphaera, Sphaerochaeta, Sphingobacterium, Spiroplasma, Tannerella, Terriglobus |
| OOD testing | Actinoalloteichus, Aeromicrobium, Agromyces, Aminobacter, Aneurinibacillus, Blastomonas, Blautia, Bosea, Brevibacterium, Cellulosimicrobium, Chryseolinea, Cryobacterium, Cystobacter, Dietzia, Ensifer, Faecalibacterium, Fictibacillus, Filimonas, Flammeovirga, Fuerstia, Gemmata, Granulosicoccus, Halioglobus, Hydrogenophaga, Labrenzia, Leclercia, Lelliottia, Lentzea, Luteitalea, Melittangium, Microbulbifer, Microvirga, Minicystis, Moorea, Mucilaginibacter, Natronolimnobius, Nitratireductor, Nitrospirillum, Nonomuraea, Olleya, Paludisphaera, Pannonibacter, Petrimonas, Planctomyces, Plantactinospora, Plesiomonas, Porphyrobacter, Rhizobacter, Rhodoplanes, Roseomonas, Roseovarius, Salinimonas, Shinella, Sphingorhabdus, Sporosarcina, Sulfitobacter, Tatumella, Tessaracoccus, Thiodictyon, Tumebacillus |

from that in the test dataset. We tune hyperparameters without exposure to the final test OOD classes. The optimal $\mu$ is 0.2 with AUROC of 0.763 in validation data and 0.727 in the test dataset. We also test if $L_2$ regularization helps for training the background model. Evaluating AUROC of OOD detection under different combinations of $\mu = [0.01, 0.05, 0.1, 0.2]$ and $\lambda = [0, 10^{-6}, 10^{-5}, 10^{-4}, 10^{-3}]$, we observe that AUROC is generally high for most of the combinations of the two hyperparameters except for some extreme cases (Table S4a), when both $\mu$ and $\lambda$ are too high ($\mu \geq 10^{-3}$ and $\lambda \geq 0.2$) such that the model fails to learn informative patterns, or both are too small ($\mu \leq 10^{-6}$ and $\lambda \leq 0.05$) such that the background model is too similar to the in-distribution specific model. The optimal combination is $\mu = 0.1$ and $\lambda = 10^{-4}$, achieving AUROCs of 0.775 in validation dataset and 0.755 in the test dataset.

Table S4: Hyperparameter tuning of mutation rate $\mu$ and $L_2$ coefficient $\lambda$ of the background model of our likelihood-raio method for genomic dataset. (a) Effects of the mutation rate $\mu$ and $L_2$ coefficient $\lambda$ on the AUROC↑ for OOD detection of genomic sequences on the validation dataset containing 2,000 in-distribution and the same number of OOD inputs. When tuning only on mutation rate $\mu$, the optimal value is $\mu = 0.2$. When tuning on both $\mu$ and $\lambda$, the optimal values are $\mu = 0.1$ and $\lambda = 10^{-4}$. (b) The same as (a) but tuning using simulated OOD inputs. The simulated OOD inputs are generated by permuting the in-distribution inputs at the mutation rate 10%. The trend of the AUROC under different combintations of hyperparameters are similar with that using real OOD inputs.

| | $\mu = 0.01$ | 0.05 | 0.1 | 0.2 | | $\mu = 0.01$ | 0.05 | 0.1 | 0.2 |
|---|---|---|---|---|---|---|---|---|---|
| $\lambda = 0$ | 0.551 | 0.664 | 0.719 | **0.763** | $\lambda = 0$ | 0.579 | 0.662 | 0.744 | **0.779** |
| $10^{-6}$ | 0.694 | 0.753 | 0.767 | 0.767 | $10^{-6}$ | 0.568 | 0.653 | 0.742 | 0.726 |
| $10^{-5}$ | 0.747 | 0.761 | 0.771 | 0.768 | $10^{-5}$ | 0.638 | 0.663 | 0.689 | 0.755 |
| $10^{-4}$ | 0.768 | 0.774 | **0.775** | 0.764 | $10^{-4}$ | 0.749 | 0.754 | **0.797** | 0.775 |
| $10^{-3}$ | 0.762 | 0.755 | 0.748 | 0.706 | $10^{-3}$ | 0.762 | 0.777 | 0.750 | 0.741 |

We further study if the hyperparameters can be tuned without using the OOD inputs. We use the perturbed in-distribution validation data as simulated OOD inputs. We choose the mutation rate as 10%, because the average identity between bacteria is estimated 96.4% at the genus level, and 90.1% at the family level (Yarza et al., 2008). Using the mutated in-distribution data to mimic OODs, we compute the likelihood-ratio for the in-distribution and the simulated OOD. The optimal ranges of the hyperparameters under which high AUROC are similar with the previous choice based on the real OODs. The optimal mutation rate when tuning without $L_2$ regularization ($\lambda = 0$), and the optimal combination of the two hyperparameters, are the same as that tuned using real OOD input.

In order to compare with the classifier-based baselines, we build a classifier using convolutional neural networks (CNNs), which are commonly used in both image and genomic sequence classification problems (Alipanahi et al., 2015; Zhou & Troyanskaya, 2015; Busia et al., 2018; Ren et al., 2018b). For genomic sequences, we feed one-hot encoded DNA sequence composed by $\{A, C, G, T\}$ into a convolutional layer, followed by a max-pooling layer and a dense layer. The output is then transformed to class probabilities using a softmax function. The number of filters, the filter size, and the number of neurons in the dense layer were tuned using the in-distribution validation dataset. This resulted in 1,000 convolutional filters of length 20 and 1,000 neurons in the dense layer. The accuracy of the classifier on the validation dataset is 0.8160. Baseline methods 6-8 are based on perturbed in-distribution inputs, so the mutation rate is a hyperparameter for tuning. We use the same validation dataset as above, and tune the mutation rate ranging from $[0.0, 0.01, 0.05, 0.1, 0.2, 0.3, 0.4, 0.5]$. For ODIN method, we tune the temperature and the input perturbation in the ranges of $[1, 5, 10, 100, 1000]$ and $[0, 0.0001, 0.001, 0.01, 0.1]$, respectively.

## C.3 Supplementary tables

Table S5 shows the minimum genetic distance between each of the OOD classes and in-distribution classes and its corresponding AUROC for OOD detection using Likelihood Ratio and classifier-based ensemble method with 20 models. We discovered that the AUROC for OOD detection is correlated with the genetic distance (Figure 5c). The Pearson Correlation Coefficient are 0.570 for Likelihood Ratio method, and 0.277 for the ensemble method. The results confirm that in general a OOD class far away from the in-distribution is easier to be detected.

Table S5: Minimum genetic distance between each of the 60 OOD classes and in-distribution classes and their corresponding AUROCs for OOD detection.

| OOD Class | Min distance | AUROC↑ (Likelihood Ratio) | AUROC↑ (Ensemble 20) |
|---|---|---|---|
| Sulfitobacter | 0.331 | 0.779 | 0.757 |
| Tumebacillus | 0.323 | 0.784 | 0.814 |
| Blautia | 0.320 | 0.708 | 0.828 |
| Roseovarius | 0.319 | 0.747 | 0.756 |
| Moorea | 0.293 | 0.904 | 0.819 |
| Natronolimnobius | 0.273 | 0.857 | 0.585 |
| Fuerstia | 0.266 | 0.887 | 0.841 |
| Chryseolinea | 0.265 | 0.887 | 0.863 |
| Faecalibacterium | 0.264 | 0.784 | 0.713 |
| Gemmata | 0.260 | 0.864 | 0.615 |
| Aneurinibacillus | 0.255 | 0.661 | 0.731 |
| Olleya | 0.253 | 0.793 | 0.820 |
| Planctomyces | 0.252 | 0.804 | 0.568 |
| Nitratireductor | 0.250 | 0.759 | 0.694 |
| Filimonas | 0.249 | 0.869 | 0.858 |
| Sphingorhabdus | 0.249 | 0.852 | 0.845 |
| Mucilaginibacter | 0.241 | 0.807 | 0.910 |
| Paludisphaera | 0.240 | 0.846 | 0.632 |
| Petrimonas | 0.240 | 0.898 | 0.878 |
| Flammeovirga | 0.230 | 0.803 | 0.836 |
| Granulosicoccus | 0.229 | 0.836 | 0.756 |
| Minicystis | 0.225 | 0.783 | 0.493 |
| Labrenzia | 0.224 | 0.720 | 0.724 |
| Microvirga | 0.224 | 0.746 | 0.725 |
| Porphyrobacter | 0.222 | 0.716 | 0.741 |
| Cellulosimicrobium | 0.217 | 0.760 | 0.601 |
| Agromyces | 0.214 | 0.704 | 0.558 |
| Melittangium | 0.209 | 0.824 | 0.692 |
| Cystobacter | 0.208 | 0.745 | 0.632 |
| Blastomonas | 0.207 | 0.782 | 0.777 |
| Pannonibacter | 0.201 | 0.778 | 0.647 |
| Ensifer | 0.201 | 0.750 | 0.758 |
| Nonomuraea | 0.197 | 0.727 | 0.527 |
| Halioglobus | 0.193 | 0.771 | 0.746 |
| Salinimonas | 0.192 | 0.796 | 0.819 |
| Microbulbifer | 0.190 | 0.790 | 0.791 |
| Roseomonas | 0.189 | 0.706 | 0.618 |
| Plantactinospora | 0.189 | 0.656 | 0.412 |
| Shinella | 0.183 | 0.651 | 0.606 |
| Aeromicrobium | 0.183 | 0.658 | 0.392 |
| Rhodoplanes | 0.179 | 0.792 | 0.730 |
| Fictibacillus | 0.179 | 0.742 | 0.738 |
| Bosea | 0.178 | 0.693 | 0.722 |
| Rhizobacter | 0.175 | 0.591 | 0.665 |
| Lentzea | 0.175 | 0.733 | 0.609 |
| Brevibacterium | 0.175 | 0.754 | 0.543 |
| Thiodictyon | 0.173 | 0.773 | 0.698 |
| Plesiomonas | 0.172 | 0.646 | 0.814 |
| Tessaracoccus | 0.170 | 0.742 | 0.533 |
| Actinoalloteichus | 0.165 | 0.706 | 0.425 |
| Sporosarcina | 0.164 | 0.802 | 0.758 |
| Aminobacter | 0.163 | 0.721 | 0.793 |
| Luteitalea | 0.162 | 0.835 | 0.700 |
| Nitrospirillum | 0.157 | 0.715 | 0.635 |
| Dietzia | 0.147 | 0.796 | 0.306 |
| Tatumella | 0.141 | 0.660 | 0.828 |
| Cryobacterium | 0.138 | 0.736 | 0.554 |
| Hydrogenophaga | 0.137 | 0.661 | 0.627 |
| Lelliottia | 0.095 | 0.535 | 0.866 |
| Leclercia | 0.094 | 0.512 | 0.807 |