[Reviews · NeurIPS 2019]

Reviewer 1



When comparing the difference in the distributions of the log likelihoods of in-distribution vs out of distribution (Fig 1a + line 94), the authors state that they largely overlap - can they be more specific, ie what is the difference in means, can you compute a p-value using e.g. Wilcoxon rank sum test, etc, rather than simply say they “largely overlap” ? The authors assume that (i) each dimension of the input space can be described as either background or semantic, and (ii) these are independent (eq 1). How true is this in practice? You could imagine e.g. a background effect of darkening an image, in which case the probability of observing a sequence depends on an interaction between the semantic and background components - similarly, the GC content of a sequence is similarly a function of the semantic component when classifying bacterial sequences. Can the authors demonstrate this assumption holds? The LLR as defined in equation (5) depends only on the semantic features -- how are these identified in practice on the test set since as the authors note z is unknown a priori? This assumption of independence between semantic and background features seems a key component of the model, and the likelihood ratio should be computed on the semantic elements alone, but it is not obvious how these are identified in practice? Or is the full feature set used and this approximate equality assumed to be true? Do the authors have an explanation for the AUROC being significantly worse than random on the Fashion MNIST dataset?

Reviewer 2



The authors were motivated to solve the problem of bacterial identification in the presence of out-of-distribution (OOD) examples: when a classifer is trained on known bacterial classes and deployed in the real world, it may erroneously classify yet unknown bacterial strains by assigning them to one of the exisiting classes with high confidence. Methods for OOD detection try to address this problem. The authors propose a novel statistic to identify OOD examples: Their method is based on taking the log-likelihood ratio (LLR) between a model trained on in-distribution data and a background model. For both models, autoregressive models are used — the background model is trained on perturbed in-distribution data (where the amount of perturbation is a hyper-parameter that needs to be tuned). Combined with the assumption that the likelihood factorises into semantic and background contributions, the statistic can be approximated as the difference in log-likelihoods under both models, effectively focusing on the semantic components only. The paper introduces a bacterial identification task, which is suggested as a benchmark for future evaluations of methods. The authors compare the performance of their method in terms of three metrics (area under ROC curve, area under precision-recall curve, and false positive rate at 80% true positive rate) on this task. The proposed LLR-based method outperforms other methods. Apart from applying their method to genomic sequences, the LLR approach is applied to image datasets. Quality This is a technically sound submission. The motivation for using the LLR statistic for OOD is clearly explained. The LLR method is compared against a representative set of baseline methods. A couple of points: - On CIFAR-10 versus SVHN (table 4) only likelihood and likelihood-ratio methods are reported, baselines are not included for comparison. This renders the comparison of methdos incomplete and the authors should include results in the revision. - I appreciated the extensive discussion of finding hyper-parameters in the supplement. However, I do not fully agree with the conclusion the authors draw: When no OOD data is available for validation and OOD data is simulated instead (table S1b), AUROC scores are high regardless of the strength of L2-regularization. The impression from table S1a is that high lambda parameters are detrimental to performance. Thus, the conclusion about plausible ranges of hyper-parameters is not the same in both settings. Would it be better to abstain from using L2 regularisation altogether in this situation, since it only seems to bring marginal gains in the best case scenario? - There is no report of variance in the metrics. To see whether the results are sensitive to e.g. the initialisation of neural networks, the authors should report errors on metrics/run baselines repeatedly. - It would be informative if runtimes of different methods (given hardware) were reported. Clarity The paper is easy to follow, and well organised. Results are cleanly presented in figures and tables. Information for reproducing results is reported. Additional points: - The authors state they "do not compare .. with other methods that do utilize OOD inputs in training.". It would be good to provide references to methods that are excluded for this reason. It might be fair to include them -- but trained on synthetic OOD data instead. Originality The authors propose a novel method for OOD that shows promising results and introduce a novel benchmark problem to the literature. Significance The idea to build test statistics on likelihood-ratios for OOD detection is interesting and opens room for developing novel techniques in future papers. Introducing new benchmarks is important, however, there is no mention of releasing code for baselines and the benchmark. The impact/adoption of the benchmark in the community will depend on the availability and ease of use of code. Without a code release, the significance of the paper for the community is reduced. The LLR statistic reaches state-of-the-art performance on the bacterial identification benchmark. It is difficult to judge, however, how this will generalize to other problem settings. Results on the CIFAR-10 versus SVHN problem are not reported with respect to the other baselines (only a comparison to likelihood is included). Typos - L279: additonal - L506 (supplement): indepedent

Reviewer 3



Originality: The idea of using likelihood ratio (LLR) with respect to a background distribution to detect OOD is novel (It has similarities to the learning of a statistical potential in the context of protein folding - https://en.wikipedia.org/wiki/Statistical_potential). Quality: The work is technically sound. Clarity: The paper is very well written and easy to understand. Significance: The experimental results are strong and give almost perfect results on Fashion-MNIST vs MNIST task. Furthermore they collect a bacterial genomic dataset (from publicly available genomes) to further evaluate their method and get significantly better results for that task as well. They also find that their OOD detection is correlated with distance to the in-distribution set giving more evidence for LLR. Given the experimental evidence and the novelty of the method, I think this is an important contribution to the stable of OOD detectors.

[Author Response · NeurIPS 2019]

We would like to thank all the reviewers for their valuable reviews and constructive feedback which will help us improve the paper. We appreciate the reviewers' very positive comments, noting e.g. that the work is *"technically sound", "very well written and easy to understand", and "an important contribution"*, the idea of LLR *"opens room for developing novel techniques in future papers", "the experiments are strong"*, and it introduces *"a novel dataset that will be of utility to the field"*, etc. Due to space constraints in the rebuttal, we focus on only the major questions and comments below, but we will address the minor comments too in the camera ready.

**R1:** *"Quantitative metrics rather than simply say they 'largely overlap'."* The AUROC of 0.630 for likelihood (Table 3) also suggests that the two histograms (Figure 1a) are barely separable. We will add it to caption of Figure 1a in revision.

*"The authors assume that (i) each dimension of the input space can be described as either background or semantic, and (ii) these are independent (Eq 1). How true is this in practice?"* Thanks for the question. We made these two assumptions in lines 118-145 to intuitively describe the high level idea of our method, but note that we later relaxed (ii) in lines 146-154. Assumption (i) is relatively mild as many input modalities consist of background plus semantic components (given appropriate resolution) e.g. background and semantic pixels in images; background noise in audio signals; key words and stop words in text; genomics studies have modelled DNA sequences with motif and background models (Reinert et al., 2009; Wan et al., 2010; Zhai et al., 2010) with successful applications (Song et al., 2013; Chan et al., 2014; Ahlgren et al., 2016). *The independence assumption in (ii) was relaxed in line 146, so our method does not rely on (ii).* Specifically, we decompose joint as product of conditional distributions using the equality $p(\boldsymbol{x}) = \prod_{d=1}^{D} p(x_d|\boldsymbol{x}_{<d})$, and use auto-regressive models, which can model any $p(\boldsymbol{x})$ given sufficient capacity to represent true conditionals $p(x_d|\boldsymbol{x}_{<d})$; $x_d$ is conditioned on all past $\boldsymbol{x}_{<d}$, so it does not introduce additional independence assumption. In Eq (4), we simply grouped likelihood terms into two groups depending on if $x_d$ belongs to background or semantic group.

*"LLR as defined in Eq 5 depends only on the semantic features–how are these identified in practice on the test set?* We do not assume knowledge of $z$ in our experiments (we will make it explicit in the text to avoid any confusion). If the modeling assumptions are correct, we expect LLR to be non-zero in the semantic part and be close to 0 on the background part. Since auto-regressive models give per-pixel likelihoods, we can validate if our hypothesis is correct by visualizing the heat map of per-pixel likelihoods and ratios. Figure 3 validates our hypothesis experimentally.

*"Do you have an explanation for AUROC being significantly worse than random on Fashion MNIST?"* Yes, we show that the proportion of zeros, i.e. the number of background pixels in an image, is highly positively correlated with the likelihood. MNIST (ood) images on average have a larger proportion of background pixels and so they get assigned higher likelihood than FashionMNIST (train). See Figures 2a-b and Figure S1,S2 in supp mat for more details.

*"...the requirement to have knowledge of the data perturbation process."* Our Bernoulli perturbations (Alg 1) make very few assumptions about data and are really simple to implement. We show that the same procedure works well for two different modalities (image and genomic data), demonstrating the generality of the methodology. We also found that training background model using just $L_2$ regularization (without any data perturbation) works effectively in some cases (cf. $\mu$=0 in Table S3). We believe other ways of training the background model can further improve performance.

**R2:** *"Additional baselines for CIFAR-10 versus SVHN."* As mentioned in line 228, our goal for image experiments was to show that LLR effectively corrects for background statistics and significantly outperforms the likelihood, and not to claim that we achieve state-of-the-art performance on these datasets. That said, we agree that having other baselines will make the table more complete. Due to limited time for rebuttal, we implemented a few classifier-based methods (using ResNet) and added the corresponding evaluation metrics to Table 1. We will provide the full table in the camera ready version. Note that our LLR is completely unsupervised whereas classifier methods require labels. Using LLR on class-conditional generative models might further improve performance.

Table 1: Additional CIFAR-10 vs SVHN results.

| | AUROC↑ | AUPRC↑ | FPR80↓ |
|---|---|---|---|
| $p(\hat{y}|\boldsymbol{x})$ | 0.914 | 0.877 | 0.084 |
| Entropy of $p(\hat{y}|\boldsymbol{x})$ | 0.924 | 0.894 | 0.132 |
| Mahalanobis distance | 0.901 | 0.920 | 0.127 |
| Ensemble 5 | 0.940 | 0.911 | 0.033 |
| Ensemble 10 | 0.943 | 0.913 | 0.024 |
| Ensemble 20 | 0.946 | 0.916 | 0.019 |
| $p(\hat{y}|\boldsymbol{x})$ with noise class | 0.922 | 0.889 | 0.063 |

*"Report errors on metrics."* Thank you for the suggestion. We will update those in the camera ready version. *"Runtimes."* The runtime of our LLR method is two times of the standard PixelCNN++ runtime.

*"Releasing code for baselines and the benchmark."* We agree that it is important to release the code and data and have been working on this since submission. The dataset and code for the genomic experiments have already been shared publicly online (for anonymity we will not share links here). All code will be publicly available by camera ready.

**R3:** Thanks for your positive comments and pointing us to the idea of learning a statistical potential in protein folding.

*"For baseline method 9 (Choi et al.), is the generative model used the same as LLR (this paper's method)?"* Yes, we do compare the two methods (LLR, WAIC) using the exact same generative model (PixelCNN++), as it allows us to compare the effectiveness of the two methods, while controlling for the generative model class.

[Meta-Review · NeurIPS 2019]

This paper applies a likelihood ratio test to detect out of distribution data, with several comparisons, and a new genomics dataset. Reviewers are supportive of this paper, and appreciated the author feedback. While ensembles perform slightly better in the CIFAR-10 vs SVHN experiment, these results will be important to include in revisions, for a balanced presentation. Great work!